# How Does the Experience of Forest Recreation Spaces in Different Seasons Affect the Physical and Mental Recovery of Users?

**DOI:** 10.3390/ijerph20032357

**Published:** 2023-01-28

**Authors:** Rui Chen, Yu Gao, Ruixin Zhang, Zhi Zhang, Weikang Zhang, Huan Meng, Tong Zhang

**Affiliations:** Landscape Planning Laboratory, Shenyang Agricultural University, No. 120 Dongling Road, Shenyang 110161, China

**Keywords:** environmental health, forest therapy, physical and mental recovery, seasonal characteristics, landscape perception evaluation

## Abstract

Background: In recent years, increasing attention has been given to the recovery effect of the forest environment on physical and mental health. Therefore, providing users with a high-quality forest landscape space is a very important research topic for forest landscape designers and forest resource managers. Main purpose: From the perspective of different seasons, this study explores the differences in landscape perceptions and physical and mental recovery of users when they experience different forest recreation spaces and the interactions between them. Methods: First, this study used virtual reality video experience and questionnaires for participants. Then, the paired-samples *t* test, one-way ANOVA and the independent-samples t test were used for statistical analysis. Finally, we also used structural equation models to analyze the relationship between landscape perception and recovery. Main results: (1) The restoration effect and perception of forest recreation spaces on people are influenced by space types and seasonal factors. (2) People’s restoration from forest environments is a gradual process from spatial cognition to emotional response. (3) The perception of the natural attributes and form of the recreation space plays a key role in the restorative effect of the environment to people, while the natural form is more important in spring than autumn. Based on the above conclusions, we suggest that the characteristic factors of the landscape environment and their different restoration effects for users in different dimensions should be considered when planning forest recreation space.

## 1. Introduction

The continuous development of the urbanization process results in a lack of access to nature, which is a phenomenon in which the relationship between man and nature is broken [1]. Under this background, with the improvement in health awareness, people have begun to consider the relationship between human development and the natural environment again, to attach importance to the support ability of natural ecosystems to human society, and to seek health from nature [2]. According to the statistics of the State Forestry Administration of China, the number of forest and grass system ecotourism tourists in China was 2.083 billion in 2021, more than half of the number of domestic tourists [3]. In this context, it is particularly important for planners to understand the mechanism by which forests affect human health and how to use the forest environment to optimize health benefits [4].

### 1.1. Forest Landscape and Physical and Mental Recovery

As an important part of the ecosystem, forests undertake many ecological functions such as improving air quality, water and soil conservation and water conservation [5]. However, in recent years, with the development of society, people have an increasing demand for the scenic recreation and health care function of forests [6]. In the past, empirical research on the health care function and public health of forests mostly focused on humans. Studies have shown that forest environmental exposure can improve cardiorespiratory function [7,8], lower blood pressure [9,10], enhance immune function [11,12], reduce the risk of cardiovascular disease [9,10,13], and improve negative emotions [10,14]. However, the effects of stress relief vary with the type of forest landscape. Forest waterscapes, especially dynamic waterscapes, have a positive impact on stress relief [15]. At the same time, a study of stand density found that the lower stand density of coniferous forest provided relaxation for the subjects, while 50% stand density of broadleaved forest produced the most stable mood [16]. The forest interior with a more unified vegetation structure greatly alleviates negative emotions [17]. Through field research on different landscape types and elements, Deng et al. found that the topographic landscape restoration effect of natural mountain forest appearance is the strongest, and the landscape elements of water, topography and plants have a significant positive impact on the restoration of human perception [9]. In different environments, the perceptual restoration level of the same landscape component varies. The optimization of environmental quality to a certain extent makes for an optimization trend in physiological or psychological indicators [18].

There have been many research achievements in forest environments and physical and mental recovery. However, the internal relationship between environmental factors and restoration effects [19] and how different landscape environmental factors affect human physical and mental health still need to be further explored [4].

### 1.2. Landscape Perception Evaluation and Physical and Mental Recovery

The process of landscape perception is that people perceive the landscape environment through their senses. Landscape perception theory is an independent theory based on environmental psychology and landscape aesthetics that studies the interaction between various elements in environmental space and tourists [20]. Two fundamental theories argue that the positive effects of contact with nature on human mental health are “restorative”: Attention Restoration Theory (ART) [21] and Psycho-evolutionary Theory [22] (also more commonly known as Stress Reduction Theory (SRT) [23]). Numerous previous studies surrounding ART and SRT have revealed the relationship between landscape aesthetics and psychophysiological experience, environmental preferences, and perceived recovery [24,25]. Perceived naturalness is largely used to describe the relationship between a landscape and its perceived degree of naturalness [26], which is an important factor in the preference for some landscapes [27]. People often prefer natural landscapes, such as mountain scenery and waterscapes [9], and their preference for landscape space is significantly influenced by landscape components (trees, shrubs and flowers) [28]. The higher the landscape quality is, the stronger the consistency of preference [29]. However, the perceived naturalness of landscape is different from that of ecology, and the perceived naturalness may have a greater impact on environmental preference than the impact of ecology [30]. Based on the study of vegetation density, dense understory vegetation space is not popular [31], and people prefer space with a strong sense of privacy and security [32]. Another study by Carrus et al. showed that the higher participants’ evaluations of natural scenes were, the higher their perceived recovery and preference scores [19]. Perceived naturalness may also play an important role in recovery [33,34].

From the above research, we can see that clarifying the environmental traits that affect individuals’ perceptions of naturalness is crucial to the exploration of environmental preference and restorative environment, but there are few relevant studies at present [35].

### 1.3. Seasonal Landscape and Physical and Mental Recovery

Visual perception is the main source of people’s perceptions and evaluations of landscapes [36]. In addition to the spatial structure, the change in the temporal structure of vegetation with the seasons is also an important factor affecting perception and preference [37]. Even evergreen plants can significantly improve landscape preference in spring and landscape restoration quality in spring, autumn and winter [38]. Research on the effect of autumn color on preference and recovery potential has found that people prefer red leaves and that autumn scenes are more likely to restore attention [39]. In addition, the seasonality of landscape elements strongly influences preferences. The flowering stage is obviously the most popular, and the color change in deciduous trees and bushes in autumn is also highly preferred [40]. The appearance of the landscape changes with the seasons. Regardless of the season, even in winter, nongreen plants are also restorative [41,42]. However, seasonal landscapes appear periodically, so people do not always pay enough attention to them [43]. Few studies have considered the dynamic changes in seasons [36,40]. The study by Agnes Peterfalvi and colleagues is the first to assess seasonal changes in the effects of the same forest environment on blood pressure, pulse, and immune parameters in the same subjects [44]. Their study demonstrated that forest walking lowers blood pressure and enhances immune function, which was even more pronounced in the late spring.

In other words, the current research on forest health restoration is mainly from the perspective of a single season [42,45,46]. However, seasonality is reflected not only in the rhythm of the natural landscape but also in the way of human life [43]. Research shows that seasonal factors account for a large proportion of tourists’ perceptions, and people’s viewing behaviors occur mainly from April to October [47,48]. Therefore, our study considered the influence of seasonal changes in the experimental design to clarify the relationship and difference between landscape perception and psychophysiological recovery in spring and autumn.

### 1.4. Application of VR Technology in Landscape Fields

As field investigations may be affected by uncontrollable factors and unexpected events, tools such as photos, virtual reality (VR) and eye trackers have rapidly developed into visual media for landscape perception and preference research [10,49,50]. Through the survey of landscape perception and preference, the effect of virtual reality is found to be more consistent with that of field surveys. Virtual reality can replace field surveys of semi open green spaces in any season and all green spaces in winter [36]. Research on physical and mental recovery shows that watching natural images can also aid in recovering from physiological stress and relaxing emotions, and virtual reality technology can be used as an alternative method to enter the natural environment for recovery [10,51,52].

Landscape experience experiments based on VR technology have been gradually recognized by scholars, and corresponding research has been carried out. In addition, the natural environment is dynamic. Simulating the natural environment and the recovery effect under highly controlled conditions may help us better understand which components of nature are conducive to stress recovery [53].

The above literature analysis shows that there are still the following problems in the current research on the relationship between tourists’ perceptions of forest recreation space and forest health restoration.

(1)The dynamic changes in seasons are seldom considered in the evaluation of health benefits, and the difference in recovery between seasons is still unclear.(2)Few studies have explored the cumulative driving effect of forest environments and individual perception factors on health benefits.(3)Related research on the effect of forest recreation spaces (FRSs) on health and the regulatory mechanisms of forest landscape perception on physical and mental recovery needs to be further developed.

Based on this, our study attempts to explore the relationship between landscape perception and people’s physiological and psychological recovery and to further explore the psychological driving mechanisms of forest therapy.

### 1.5. Research Purpose and Hypothesis

In summary, our study explores the differences between individual landscape perceptions (perceived naturalness, environmental preference, perceived restorativeness) and psychophysiological recovery and the interaction between them from the perspective of different seasons. We analyzed the interaction mechanism between users’ landscape perceptions and physical and mental recovery in forest recreation spaces (FRSs) to enrich the research in the field of perception, provide a theoretical basis for creating high-quality forest therapy activity space, and then enhance the ecological recreation value of forests in the future.

The following hypotheses are proposed in this study:

**Hypothesis** **1** **(H1).**
*There are differences in people’s landscape perceptions and physical and mental recovery across seasons and spaces.*


**Hypothesis** **2** **(H2).**
*People’s perceptions of forest recreation spaces can promote physical and mental recovery.*


**Hypothesis** **3** **(H3).**
*The functional process of landscape perception on physical and mental recovery varies across seasons and space.*


## 2. Materials and Methods

### 2.1. Study Sites

Considering the travel distance of people, this study selects the forest recreation areas (forest parks, scenic spots and scenic recreation forests) within the city suburbs (within 60–100 km from the city center) as the research object. Based on previous studies [49,54], this study selected three landscape types as sample plots: in-forest landscapes, water landscapes and lookout landscapes (scenic viewpoint). According to the types of vegetation composition, the in-forest landscapes were divided into coniferous forest landscapes, broadleaved forest landscapes and mixed forest landscapes. The water landscapes were likewise divided into dynamic water landscapes and static water landscapes. According to the distribution of natural forest resources in Liaoning Province, after investigation and exploration, from the perspective of spatial representation and universality, Hemu National Forest Park, Greenstone Valley National Forest Park, the Liberated Forest in Caohekou and Phoenix Mountain National Scenic Area were selected as research sample plots. Twenty experts in the field of landscape architecture evaluated the physical characteristics of 24 sites by viewing panoramic photos and ultimately selected the above six FRSs as experimental materials (Figure 1).

### 2.2. Experimental Materials

To maximize the sense of presence at the scene, the study used 5-min VR videos as visual stimulation material. VR videos were shot with a 360 panorama camera (Insta360 ONE R (Twin)) in clear and windless weather. When shooting, the camera bracket was set at a height of 1.4 m from the human point of view [15]. The autumn experimental materials were shot from 9:00 a.m. to 11:00 a.m. in July 2019, and the spring experimental materials were shot from 9:00 a.m. to 11:00 a.m. in May 2021 (Table 1).

The dynamic water landscape and lookout landscape are located in the Phoenix Mountain Scenic Area. The static water landscape is located in Greenstone Valley National Forest Park. The broadleaved forest landscape is located in Hemu National Forest Park. The coniferous forest landscape and mixed forest landscape are located in Liberation Forest.

### 2.3. Participants

Because landscape perception is influenced by people’s complex social attributes, it is difficult to achieve absolute unity of various indicators of the sample, resulting in great heterogeneity of the conclusions [55]. To make the conclusion more universal, the study selected college students with similar age and experience and at the same level of knowledge about the landscape as subjects to ensure the homogeneity of the sample population [56]. The participants were fully informed of the purpose and procedure of the study and were told to avoid strenuous physical activities, smoking and drinking alcohol one day before the experiment. Our procedures and requirements for recruiting these volunteers fully complied with the ethical standards of the College of Forestry, Shenyang Agricultural University, China (CF-EC-2022-003).

### 2.4. Evaluation Scales and Measurement Methods

#### 2.4.1. Physiological Measures

Biofeedback measurement was used in this study. Blood pressure (systolic blood pressure, SBP; diastolic blood pressure, DBP) and heart rate (HR) were selected as indexes to measure the physiological changes in college students.

Blood pressure (BP) and HR are the most commonly used indicators to measure the health of the cardiovascular system, reflecting the activity of the autonomic nervous system, including the sympathetic nervous system and parasympathetic nervous system. To ensure the stability of the experimental data, an Omron electronic sphygmomanometer (HEM-7136, Japan) was used to continuously measure SBP, DBP and HR three times, and the average value was calculated.

#### 2.4.2. Psychological Measures

For psychological states, the Mood Check List-Short form.2 (MCL-S.2) was selected to assess subjective emotional changes. The MCL-S.2 is an emotional assessment tool, that research has shown has credibility and plausibility [57,58]. The scale was compiled with a seven-point Likert scale, including 12 items and three emotional actors: pleasantness (P), relaxation (R) and anxiety (A).

#### 2.4.3. Landscape Perception Evaluation Scales

In this study, the questionnaire on landscape perception included the perceived naturalness scale (PNS), the environment preference scale (EPS) and the perceived restorativeness scale (PRS). Among them, the EPS and the PRS use existing mature scales. EPS includes four dimensions: coherence, legibility, complexity and mystery. PRS includes four dimensions: being away, extent, fascination and compatibility. The PNS was adapted from a previous research questionnaire [35]; a total of 30 people were invited to participate, including 10 landscape architecture teachers, 10 landscape practitioners, and 10 nonprofessionals. Combined with the situation of the sample plots and the purpose of the study, we finally put forward 12 items in three dimensions to evaluate the perceived naturalness of forest landscapes. The three evaluation dimensions are perception of natural attributes (i.e., natural attributes, Cronbach’s α = 0.765, CR = 0.808, AVE = 0.521), perception of natural space (i.e., natural space, Cronbach’s α = 0.769, CR = 0.822, AVE = 0.538) and perception of natural form (i.e., natural form, Cronbach’s α = 0.813, CR = 0.838, AVE = 0.564) (Table 2). All the above scales were compiled with a seven-point Likert score.

### 2.5. Experimental Design

To ensure the independence of the experiment, the subjects were randomly assigned to a sample for the VR experience. To ensure that the subjects’ physical and mental state were in the stress condition before the experiment, the study used the Trier Social Stress Test (TSST) to induce acute stress in the subjects. In this study, the TSST consisted of a 3-min public presentation and a 2-min oral arithmetic task under noise [15,50,53]. The experiment was conducted from 8:00 to 12:00 on 15 October to 28 December 2021. The specific experimental process is shown in Figure 2.

### 2.6. Statistical Analysis

We used SPSS 22.0, Amos 24.0 and Smart-PLS 3 for data analysis. First, we performed the reliability and validity tests for all questionnaires, and the results indicated that all data were plausible (Cronbach’s α > 0.7, CR > 0.6, AVE > 0.5) except for the natural space (Cronbach’s α = 0.711, CR = 0.685, AVE = 0.359). Skewness and kurtosis were calculated to check the normality of the variables of this study. The variable skewness ranged from −1.646~1.516, and the kurtosis range was from −0.789~6.900, which satisfied skewness <3 and kurtosis <10, indicating that all variables were normally distributed [59]. After that, we conducted an in-depth analysis of the data. The specific analysis steps were as follows:(1)A paired sample T test in SPSS 22.0 was used to test the changes in physiological and psychological indexes before and after watching forest landscape VR videos.(2)One-way ANOVA in SPSS 22.0 was used to compare the differences in physiological changes, psychological changes and landscape perception of users across types of FRSs in spring and autumn.(3)The independent-samples t test in SPSS 22.0 was used to compare the physiological changes, psychological changes and landscape perception differences in users across FRSs in different seasons.(4)The relationship between physiological and psychological changes and landscape perception was analyzed by a structural equation model accompanied with the maximum likelihood estimator (ML-SEM) in Amos 24.0. The fitting index of the measurement model suggested by Mulaik et al. [60] was used to test the structural equation. Partial least squares (PLS-SEM) in Smart-PLS 3, which is more suitable for small sample sizes, was used to construct structural equations for each space [61]. Bootstrapping was performed to verify the mediating effect.

## 3. Results

In this study, 536 people were recruited voluntarily through on-site invitations and online invitations, and the final valid dataset comprised data from 520 people. Among them, there were 251 males and 269 females who came from landscape architecture, forestry, agriculture, plant protection, water conservancy engineering and other majors. The participants were between 17 and 24 years old and in good health, and their myopia was less than 800 degrees. The demographic parameters of the participants are shown in Table 3.

### 3.1. Physical and Mental Recovery Effect of FRSs

#### 3.1.1. Changes in Indicators before and after FRSs Experience

Regardless of space type and seasonal change, viewing natural landscapes has a positive impact on the physiological and psychological indicators of subjects.

The data analysis shows that BP and HR decreased significantly (*p* < 0.001) after watching landscape VR videos under pressure. SBP and DBP decreased by 14.344 mmHg (SD = 6.928, 95% CI [13.747, 14.941]) and 11.307 mmHg (SD = 6.221, 95% CI [10.771, 11.842]) on average, respectively, and HR decreased by 13.629 bpm (SD = 7.860, 95% CI [12.952, 14.307]). The results of the MCL-S.2 analysis show that watching landscape VR videos has a significant positive impact on all scores (*p* < 0.001). The positive emotions of P (M = 0.916, SD = 1.152, 95% CI [0.817, 1.016]) and R (M = 1.280, SD = 1.320, 95% CI [1.166, 1.394]) increased significantly, while the negative emotions of A (M = 1.037, SD = 1.132, 95% CI [0.939, 1.134]) decreased significantly.

#### 3.1.2. Differences in the Physical and Mental Recovery of Each Space in the Same Season

In terms of physiological recovery, the six spaces had similar effects on the physiological indexes of subjects in the two seasons. Among them, the restoration values of water landscapes (dynamic water and static water landscape) (*p* < 0.01) and lookout landscape (*p* < 0.05) are significantly higher than those of in-forest landscapes (coniferous forest, broadleaved forest and mixed forest landscape) (Figure 3(A-1,A-2)).

In terms of psychological recovery, the measurement results of MCL-S.2 show no significant differences in P, R, or A in each space between spring and autumn (Figure 3(B-1,B-2)). However, through the numerical comparison, we can still find that regardless of the season, water and lookout landscapes had a better influence on mood than in-forest landscapes. In addition, the mixed forest landscape in spring had a more positive influence on mood recovery.

#### 3.1.3. Seasonal Differences in Physical and Mental Recovery across Recreation Spaces

In terms of physiological recovery, through seasonal comparison, we found that the recovery effects of SBP (t = −3.656, *p* = 0.0004), DBP (t = −3.815, *p* = 0.0003) and HR (t = −2.885, *p* = 0.0050) in the dynamic water landscape were significantly higher in spring than in autumn. The mixed forest landscape in autumn (t = 2.737, *p* = 0.0075) promoted the recovery of SBP, while the broadleaved forest landscape in autumn promoted the recovery of DBP (t = 2.422, *p* = 0.0176) (Figure 4A).

Although no significant difference was found in the comparison of indexes in other spaces, we found that the recovery effect of BP was higher in spring for water and lookout landscapes than in autumn, while the recovery effect of BP was higher in autumn for in-forest landscapes. All landscapes except for the dynamic water landscape showed a better recovery effect of HR.

In terms of psychological recovery, although the independent-samples t test only showed a significant difference in the score of R at the broadleaved forest landscape (t = −2.110, *p* = 0.0378) (Figure 4B), we still found that the score of A in the dynamic water landscape (spring: M = −1.064, SD = 1.281, 95% CI [−1.458, −0.670]; autumn: M = −0.921, SD = 1.286, 95% CI [−1.311, −0.530]) and the score of *p* in the static water landscape (spring: M = 1.165, SD = 1.263, 95% CI [0.781, 1.549]; autumn: M = 0.983, SD = 1.065, 95% CI [0.659, 1.037]) were slightly better in spring than in autumn by comparing the scores of the MCL-S.2 scale. Except for the mixed forest landscape, the scores of other items were better in autumn than in spring in the other five spaces. In the mixed forest landscape, the MCL-S.2 scores in spring were better than those in autumn.

### 3.2. Analysis of the Landscape Perception Questionnaires

#### 3.2.1. Differences in Landscape Perception across Spaces in the Same Season

In spring, the landscape perception evaluations were significantly different in all the dimensions except for the mystery of environmental preference (F(5, 253) = 1.236, *p* = 0.293) and the being away of perceived restorativeness (F(5, 117.571) = 1.035, *p* = 0.400) (Figure 5).

The data analysis of each recreation space in autumn shows significant differences in the three dimensions of perceived naturalness (natural attributes: F(5, 255) = 5.221, *p* = 0.0001; natural space: F(5, 255) = 3.937, *p* = 0.0019; natural form: F(5, 255) = 6.267, *p* = 0.00002). There were significant differences in the coherence (F(5, 118.530) = 2.855, *p* = 0.0180) and complexity (F(5, 255) = 3.273, *p* = 0.0070) of environmental preference and the fascination (F(5, 255) = 2.863, *p* = 0.0155) and extent (F(5, 255) = 2.912, *p* = 0.0141) of perceived restorativeness (Figure 5).

On the whole, regardless of the seasonal characteristics, the perception evaluations of water and lookout landscapes were generally better than those of in-forest landscapes.

#### 3.2.2. Seasonal Differences in Landscape Perception across Recreation Spaces

In general, the evaluation of the perceived naturalness of all recreation spaces was higher in autumn, the evaluation of the environmental preference of waterscapes was higher in spring, and the dynamic water landscape and mixed forest landscape in spring resulted in higher perceived restorativeness evaluations (Figure 5).

At the same time, by analyzing the influence of seasonal changes on people’s landscape perception evaluations, we found that the landscape perception of in-forest landscapes across seasons was only significantly different in the natural space perception in the mixed forest (t = −2.387, *p* = 0.0192). However, seasonal changes had different effects on landscape perception in water and lookout landscapes. This is reflected in the four dimensions of natural attributes (static water: t = −2.390, *p* = 0.0190), natural space (lookout: t = 2.268, *p* = 0.0259), legibility (static water: t = 2.116, *p* = 0.0373) and extent (dynamic water: t = −2.477, *p* = 0.0152; lookout: t = 3.188, *p* = 0.0020). Furthermore, there was no significant seasonal difference (*p* > 0.05) between coniferous forest and broadleaved forest in each evaluation dimension (Figure 5).

### 3.3. Relationship between Landscape Perception and Physical and Mental Recovery

#### 3.3.1. Relationship between Landscape Perception and Physical and Mental Recovery in FRSs

The maximum likelihood method was used for parameter estimation. After correction, the measurement and research model fit well, and the final ML-SEM is shown in Figure 6.

According to the standardized structural model, landscape perception has a significant positive impact on physical and mental recovery. The restorative experience can be obtained mainly through three paths: “natural attributes → natural form → physiological recovery (SBP and DBP)”, “natural attributes → environmental preference → perceived restorativeness → psychological recovery → physiological recovery (SBP and DBP)” and “natural attributes → natural form → environmental preference → perceived restorativeness → psychological recovery → physiological recovery (SBP and DBP)”. Among them, perceived naturalness has a direct impact on people’s physiological recovery and has an indirect impact on psychological recovery through environmental preference and perceived restorativeness. From the analysis of natural attributes and natural form in perceived naturalness, we find that natural attributes can not only directly affect people’s environmental preferences but also indirectly promote environmental preferences and physiological recovery through natural form.

#### 3.3.2. Relationship between Landscape Perception and Physical and Mental Recovery across Seasonal Characteristics and Space Types

The spring and autumn samples are used to construct the ML-SEM, and each space sample is used to construct the PLS-SEM. All fitting indexes of all models are in line with standards, and the standardized path coefficients of each season are shown in Figure 7 (see Appendix A for each space model diagram).

From the structural equations of the two seasons, the landscapes had a similar regulatory mechanism on the physical and mental recovery of people in spring and autumn. Perceived naturalness had indirect effects on psychological recovery through environmental preference and perceived restorativeness. The difference is that for physiological regulation, natural attributes in autumn can directly stimulate people to produce physiological recovery, while in spring, natural attributes indirectly affect physiological recovery through natural form, and natural form plays a mediating role. For the regulation of mood in autumn, the indirect effect of natural attributes on environmental preference through natural form is also less than the direct effect of natural attributes on environmental preference.

Through the structural equation of the 12 spaces, we found that natural form played a completely intermediary role between natural attributes and environmental preference in 4 spaces in spring (static water, lookout, broadleaved forest and mixed forest) as well as 1 in autumn (lookout). It played a partially intermediary role in 2 spaces in spring (dynamic water and coniferous forest) and 4 in autumn (dynamic water, static water, broadleaved forest and mixed forest). Moreover, the mediating effect of natural form was not found in the autumn coniferous forest landscape. It can be seen that the natural form of recreation space is more important to FRSs in spring. At the same time, we found that viewing recreation space will not only directly promote physiological recovery but also provide feedback to physiology through more complex psychological paths. Based on this, we speculate that there is an interaction between the physiological and psychological recovery caused by landscape stimulation.

## 4. Discussion

### 4.1. Landscape Perception and Restoration Effects in FRSs Are Influenced by Space Types and Seasonal Characteristics

#### 4.1.1. Water and Lookout Landscapes Have Obtained Higher Perceptual Evaluation and Physical and Mental Recovery

Our study found that different types of FRSs made significant differences in participants’ physiological recovery (Figure 3). There were also significant differences in the perception of recreation space (Figure 5). Overall, the landscape perception evaluations and physical and mental recovery effects of waterscapes and lookout landscapes were higher than those of in-forest landscapes. To some extent, this verifies H1 from the perspective of space.

The finding of this study is compatible with SRT that the characteristic feature of ‘naturalness’ creates a positive restoration effect on humans. According to the SRT, humans are biologically connected to safe and natural environments that possess trees, water, and other vegetation for immediate positive responses [22,23]. Previous studies have shown that waterscapes have a positive effect on relieving stress [15]. In forests, ‘*Overlooking*’ can increase parasympathetic activity and decrease sympathetic activity more than walking [62]. Our conclusions are supported by previous studies. This may be because people prefer blue space and partially open green space [51]. Moreover, vegetation density also affects people’s perception preferences and physical and mental recovery; too high a density will bring people a sense of unease [32], and medium density can reduce stress more than low or high density [50]. The sense of openness brought by the waterscape and lookout landscape may be due to their higher perception evaluations and restoration effects [63]. The Prospect Refuge Theory argues that landscapes with prospects and a panoramic view are preferred by people [64]. Although some previous studies found differences in psychological recovery across natural environment types [7,10,65], this difference was not significant in our experiment. We suspect that this may be because the special period of the epidemic has limited people’s daily lives and caused great psychological stress [66,67], so any natural environment will produce a very high recovery effect. It is worth noting that worse mental states may respond more significantly to the natural environment, and each individual mental state should be considered in future studies.

#### 4.1.2. Seasonal Changes in FRSs Can Better Promote Perception and Recovery

Through seasonal comparison, we found that the physiological recovery of the dynamic water landscape in spring was significantly better than that in autumn, and there were significant differences between spring and autumn in other physiological indexes for the other spaces (Figure 4). For the analysis of landscape perception, there were also differences in different indicators in the waterscapes, lookout and mixed forest landscape (Figure 5). This result verifies H1 from the perspective of seasons, although we still have not found a seasonal difference in psychological recovery.

As in previous studies, whether in spring or autumn, natural landscapes have a restorative effect on people [46,65,68]. However, through the seasonal comparison, we further find that waterscapes and lookout landscapes result in higher physiological recovery in spring. This is similar to the findings of Peterfalvi et al. [44], who found that walking in forests in late spring has more obvious effects on lowering blood pressure and enhancing immune function than walking in winter. Of course, this may also be caused by seasonal deviation. That is, there is an interaction between the season during which photos are taken and the season in which photos are viewed. When the evaluation is made in late summer, the preference for autumn leaf photos increases [69]. This suggests a forward preference for the coming season. While our experiment was conducted in winter, and participants who were tired of winter may have been awakened by the stimulus related to the coming spring.

The color of vegetation is an important reason for the difference in landscape perceptions, and the color of vegetation varies with the seasons [70]. However, repeated space elements and uniformity of colors make people bored, thus resulting in visual unattractiveness to humans [71]. This may be the reason why the physiological recovery in spring was not as good as that in autumn in our study. The research of Półrolnicza et al. [56] shows that there is no significant difference in landscape perceptions across seasons, which is consistent with our research results. However, through numerical comparison, we can still find that people’s perception evaluations and psychological recovery in autumn show a high trend. This may be because the richer colors in autumn landscapes and the diversity of seasonal landscape changes increase the aesthetic landscape quality and thus strongly affect the evaluations of perceived preference, providing people with positive physiological and psychological responses [40,45].

However, different from other spaces, the mixed forest landscape resulted in higher psychological recovery in spring. At the same time, we found that insect elements such as butterflies and bees existed in the experimental video of combined forest landscape in spring, which made it receive positive evaluation. This is not surprising because humans are usually more interested in animals than plants [71]. Other studies have found that the species richness of birds, butterflies, bees and other insects perceived by respondents in a green space environment has a certain correlation with psychological recovery [1,72].

That is, the study showed that waterscapes and lookout landscapes result in higher physiological recovery in spring, and in-forest landscapes result in higher physiological recovery in autumn. Moreover, seasonal changes in forest landscapes and the perceived diversity of animals and plants can better promote psychological recovery.

### 4.2. The Impact of the Forest Environment on Human Physical and Mental Recovery Is a Gradual Process

#### 4.2.1. Perceptual Evaluation of FRSs Induces Physical and Mental Recovery

Previous studies have shown that the naturalness degree of perception is associated with increased perceived restoration [25,73], and this increase in perceived restoration is associated with increased well-being [1]. At the same time, the bivariate correlation among perceived naturalness, environmental preference and perceived restorativeness score is very strong [19]. We not only verified this relationship but also deeply explored the mechanism of action among them (Figure 6).

We found that landscape perceived naturalness acted on psychological recovery through environmental preference and perceived restorativeness. This partly confirms H2. From the perspective of environmental psychology, it is a gradual process of psychological perception for participants to obtain psychological recovery from the natural environment. The process can be explained by temporal self-regulation theory [74], which means that people’s evaluations generated by environmental stimuli will promote the generation of emotion and further affect individual behavior (Figure 8).

However, we further found that the natural perception of the landscape environment can not only directly promote physiological recovery but also indirectly affect physiological recovery through complex psychological paths. The SRT posits that in natural environments not only a sense of restoration with emotions is experienced consciously but also unconscious physiological reactions are triggered that provide rapid short-term recovery from stress [22,23]. We suspect that this phenomenon may occur because the environment triggers the physiological response again after mobilizing positive emotions. There is interaction between the two. This may also be due to the existence of competitive mediation [75].

#### 4.2.2. Effects of Attributes and Form of FRSs on Physical and Mental Recovery (Spring: Attributes + Form; Autumn: Attributes)

To further clarify the mechanistic differences between spring and autumn, we performed a more detailed analysis. By comparing the regulatory paths of landscape perception to physiology, we find that natural attributes directly affect physiological recovery in autumn, while natural form plays an intermediary role between natural attributes and physiological recovery in spring. Regarding the psychological regulation of landscape perception, spring and autumn have similar action paths. Natural attributes directly affect environmental preferences and indirectly affect environmental preferences through natural form (Figure 7). The natural form still mediates the relationship between natural attributes and environmental preferences in both seasons. However, the indirect effect of natural form is less than the direct effect of natural attributes in autumn FRSs. Through the analysis of each autumn space, the complete mediation effect of natural form is found only in the autumn lookout landscapes (Appendix A). This finding confirms H3.

Seasons affect perceptual preference mainly because the biological characteristics of plants change the appearance and ecological characteristics of plant communities with the change in seasons, such as the change in color, shape and biodiversity, thus affecting people’s visual perceptions and psychological response [56,70]. Previous studies have shown that the color of vegetation is an important reason for the difference in landscape perception [70] and that species richness can influence preference evaluations and improve well-being [1,40]. Based on this, we speculate that the color changes brought by autumn increase people’s perception diversity of space elements, so natural attributes can directly affect physical and mental recovery. In spring, because of its uniform color, species diversity cannot be well identified by color. Therefore, natural attributes need to work together with form to restore the body and mind. The autumn lookout landscape is a bird’s eye perspective, rather than the ground-level visual perspective that individuals actually have of the green spaces around them. Color alone cannot provide a good understanding of the space, so we also need the help of form to restore our body and mind.

#### 4.2.3. Thoughts on the Influence Mechanism of FRSs on Physical and Mental Recovery

Although our research reveals the influence mechanism of landscape perception on physical and mental recovery, the dimension of natural space in perceived naturalness is not included. A significant correlation between the three dimensions of perceived naturalness and self-rated restoration was shown by Liu et al. [35]. Joye and van den Berg [76] pointed out that landscape preference is not the direct cause of physical and mental recovery. However, psychological research has shown that the initial emotional response of preference is triggered immediately after receiving environmental stimulation, and preference is the product of perception [77]. Moreover, our study found that the perception of landscape naturalness can simultaneously directly affect physiological recovery and psychological recovery in a complex way. There may be an interaction between the two.

Based on our analysis, we believe that the restoration effect of forests on people is a complex comprehensive result influenced by space type, seasonal characteristics and perceptual evaluations. At the same time, various factors (space types, seasonal characteristics and perceptual evaluations) also have mutual constraints and promotion relationships, which indirectly affect the physical and mental restoration effects of forests (Figure 9).

### 4.3. Limitations

Our research starts from the perception of landscape naturalness and quantitatively analyzes the recovery effect of landscape perception on physiology and psychology. However, it still has some limitations.

(1)Although the PNS in this study met the requirements in terms of validity and reliability, there may still be problems such as insufficient evaluation dimensions and deviation in the selection of indexes. In future research, we hope to develop a more comprehensive questionnaire to evaluate perceived naturalness by combining objective indicators (e.g., visible green index, vegetation structures, tree age, species diversity, environmental meteorological index).(2)This study discussed only the characteristics and differences in perception evaluation and restoration benefit in spring and autumn. The physical and mental recovery effects of forest space in summer and winter in northern China need to be further explored.(3)Landscape perception is a multisensory dimension and should not be limited to vision. Future research should further explore sensory preferences such as hearing and smell and combine them with visual landscape preferences to comprehensively explore human perception.(4)Demographic characteristics (e.g., gender, occupation, age, educational background, professional knowledge) also have a certain impact on the results of landscape perceptions and preferences. In the future, more levels of samples still need to be enriched to explore the differences in different demographic characteristics.

## 5. Conclusions

Based on different space types, this study further explored the differences in FRSs restoration across seasons and tried to use a structural equation model to preliminarily reveal the regulatory mechanism of landscape perception on physical and mental recovery.

The specific conclusions are as follows:(1)The restoration effect and perception of FRSs on people are influenced by space types and seasonal factors.(2)People’s restoration from the forest environment is a gradual process from spatial cognition to emotional response, and there may be interaction between physiological recovery and psychological recovery.(3)The natural attributes and form of recreation space play a key role in the restoration of the environment to people, and the natural form is more important to FRSs in spring than in autumn.

These findings extend and deepen the psychological driving mechanism of forest therapy and enrich research in the field of forest landscape perception. This study provides a new perspective for understanding the effect of forest recreation environments on human health and provides theoretical support for the planning and design of forest therapy spaces.

Based on the above conclusions, we suggest the following:(1)We found not only the important role of natural attributes of forest space but also that natural form is important for promoting human health. Therefore, it is necessary to optimize the environmental quality of forest therapy bases from the aspects of forest structure and environmental construction. In addition to paying attention to the landscape elements themselves, it is also necessary to create a continuous, hierarchical and varied forest environment. It is important to enhance the natural perception of people in recreational mode, encouraging them to have a preference for the environment to arouse perceptual restorativeness and achieve the ultimate goal of physical and mental recovery.(2)This study showed that meeting with insects or small animals may bring unexpected surprises to recreationists, thus enhancing their interest and producing a better recovery effect. Therefore, in the design of recreation space, the configuration of fruit-source plants and honey-source plants can be considered to improve the presence of insects and small animals for the benefit of recreationists.(3)This study also found that forest therapy activities in water and lookout landscapes in spring, and forest therapy activities in autumn will produce a better recovery effect. Considering the different restoration effects brought by the characteristic factors of the landscape environment, it is suggested to plan more forest rehabilitation activities in water and lookout landscapes.

## Figures and Tables

**Figure 1 ijerph-20-02357-f001:**
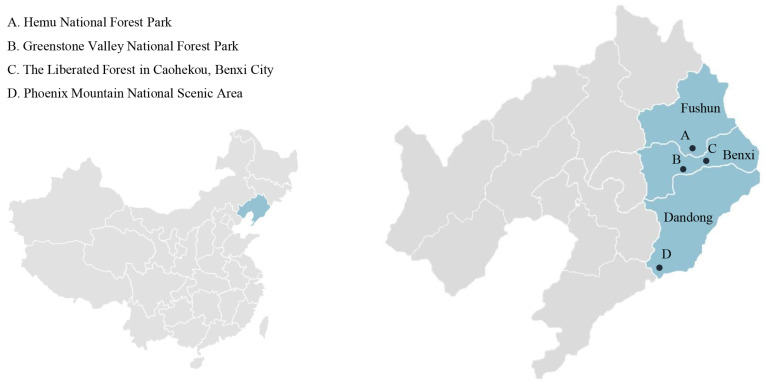
Locations of the study sites. A: Hemu National Forest Park has a total area of 13.68 square kilometers. The environmental quality in the park is good, and the vegetation coverage rate is high. There are natural secondary forests and plantations. B: Greenstone Valley National Forest Park covers an area of approximately 20 square kilometers, with undulating terrain and diverse vegetation. It is a typical mountain-type forest park. C: The Liberation Forest in Caohekou, Benxi City, has 0.22 square kilometers of artificially planted Korean pine forest; it is the first artificially built Korean pine forest in China. D: Phoenix Mountain National Scenic Area covers an area of approximately 216 square kilometers, with high mountains and forests, waterfalls and springs, and rich dynamic water space. It is a typical scenic area in eastern Liaoning.

**Figure 2 ijerph-20-02357-f002:**
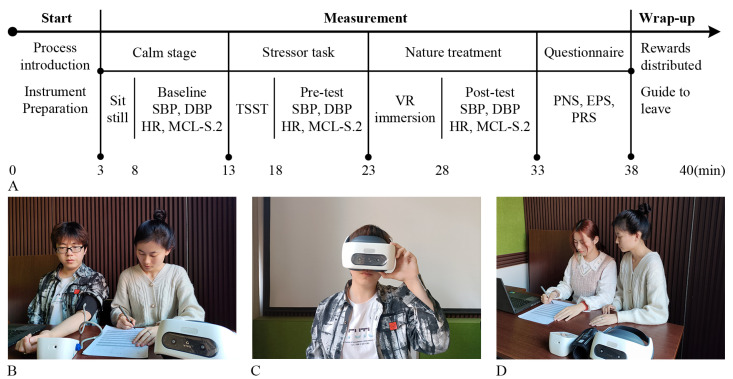
Schematic diagram of the experiment. (**A**) Experimental flow chart. (**B**) Measurement of physiological and psychological indicators. (**C**) Landscape VR experience for 5 min. (**D**) Completion of landscape perception evaluation questionnaires. SBP: systolic blood pressure; DBP: diastolic blood pressure; HR: heart rate; MCL-S.2: mood check list-short form.2; TSST: trier social stress test; VR: virtual reality; PNS: perceived naturalness scale; EPS: environment preference scale; PRS: perceived restorativeness scale. This figure was created by Rui Chen.

**Figure 3 ijerph-20-02357-f003:**
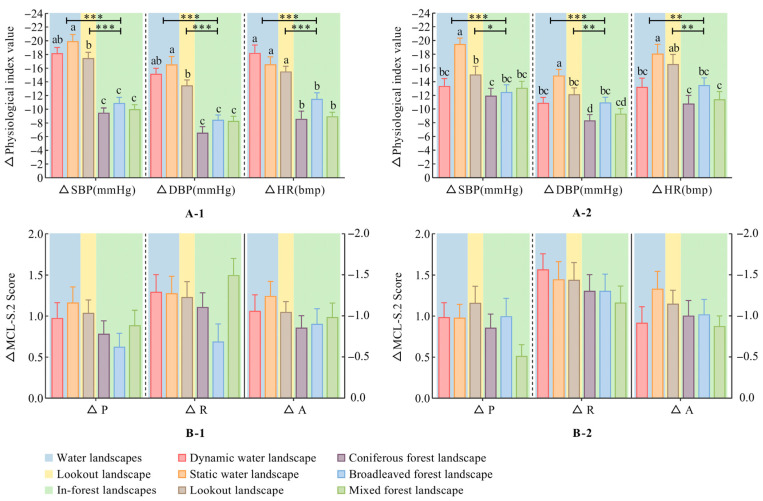
Comparison of the changes in physiological and psychological indexes of recreation spaces in the same season. (**A-1**) Spring physiological indexes. (**A-2**) Autumn physiological indexes. (**B-1**) Spring psychological indexes. (**B-2**) Autumn psychological indexes. Mean ± SE; “*”: significant differences among the three landscape types, * *p* < 0.05, ** *p* < 0.01, *** *p* < 0.001; “Lowercase letters”: significant differences (*p* < 0.05) among the six spaces; one-way ANOVA. △D = post–pre. SBP: systolic blood pressure; DBP: diastolic blood pressure; HR: heart rate; P: pleasantness; R: relaxation; A: anxiety. This figure was created by Rui Chen.

**Figure 4 ijerph-20-02357-f004:**
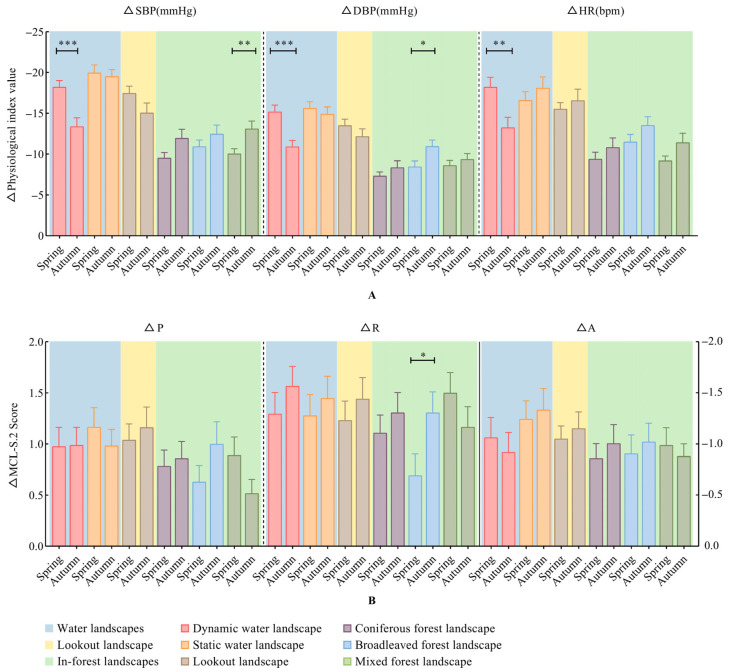
Seasonal differences in physiological and psychological index variation across recreation spaces. (**A**) Physiological indexes. (**B**) Psychological indexes. Mean ± SE; * *p* < 0.05; ** *p* < 0.01; *** *p* < 0.001; Independent-samples *t* test. ΔD = post–pre. SBP: systolic blood pressure; DBP: diastolic blood pressure; HR: heart rate; P: pleasantness; R: relaxation; A: anxiety. This figure was created by Rui Chen.

**Figure 5 ijerph-20-02357-f005:**
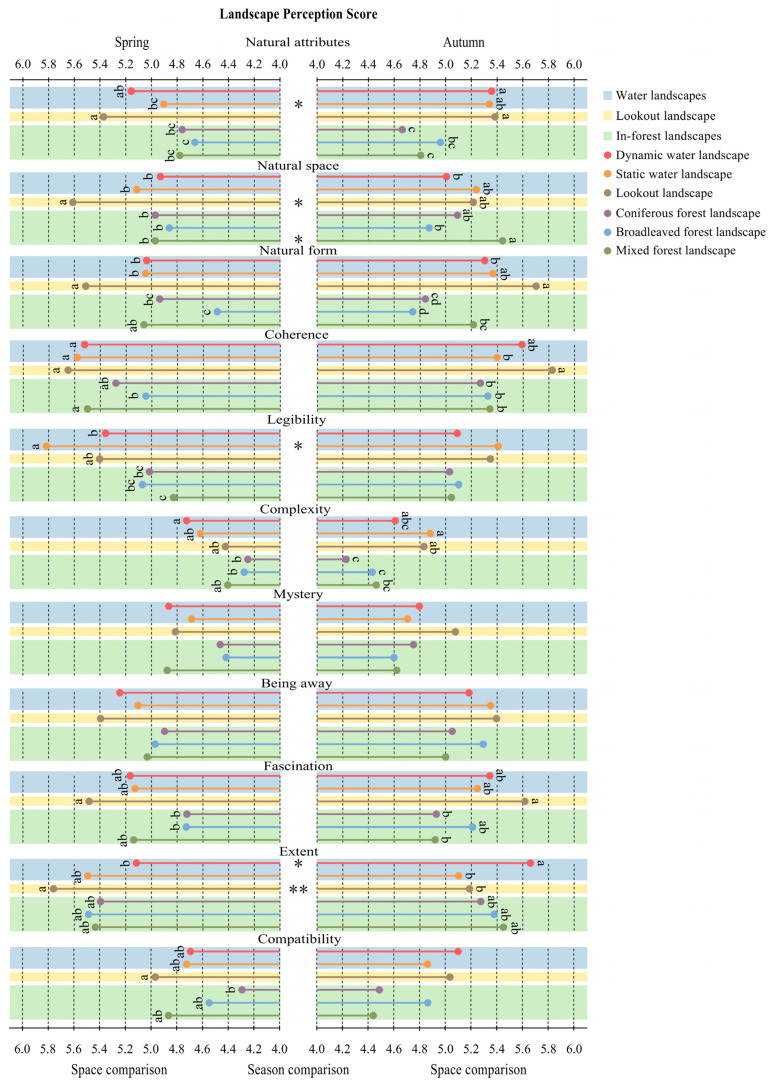
Comparison of mean scores of the Perceived Naturalness Scale, Environmental Preference Scale and Perceived Restorativeness Scale. Perceived naturalness scale: natural attributes, natural space and natural form. Environmental preference scale: coherence, legibility, complexity and mystery. Perceived restorativeness scale: being away, extent, fascination and compatibility. “Lowercase letters”: significant differences (*p* < 0.05) among the six spaces. “*”: significant differences between the spring and autumn; * *p* < 0.05; ** *p* < 0.01. Landscape space comparison: one-way ANOVA; Seasonal comparison: independent-samples t test. This figure was created by Rui Chen.

**Figure 6 ijerph-20-02357-f006:**
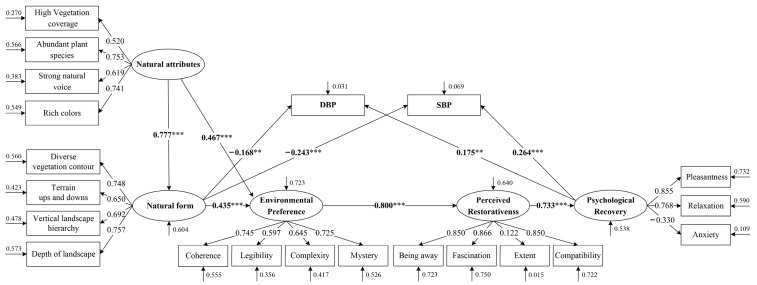
The final associations among the study variables. Fit indexes: Chi-square/df = 1.18, GFI = 0.97, AGFI = 0.96, RMSEA = 0.02, IFI = 0.99, NFI = 0.97, CFI = 0.99, TLI = 0.99, RFI = 0.97. The model fits well. ** *p* < 0.01; *** *p* < 0.001. SBP: systolic blood pressure; DBP: diastolic blood pressure. This figure was created by Rui Chen.

**Figure 7 ijerph-20-02357-f007:**
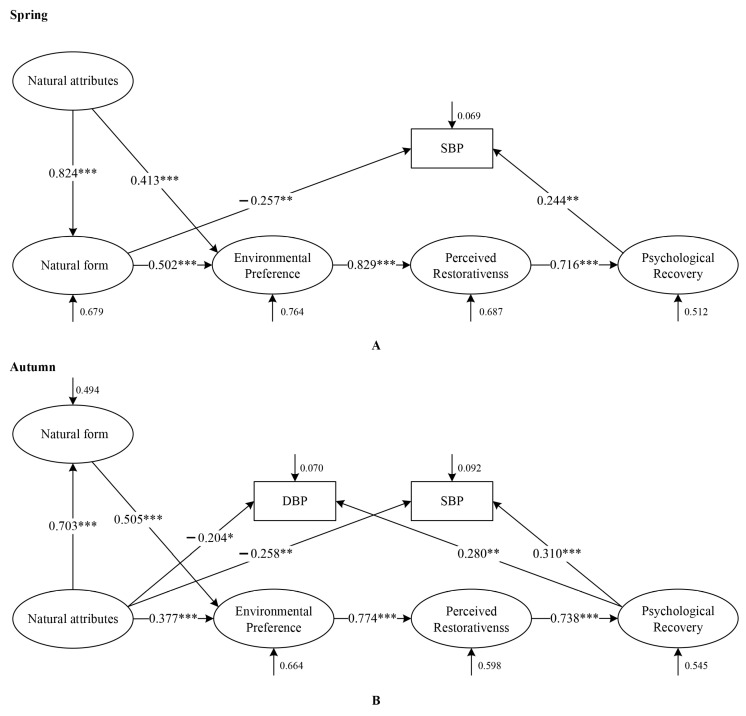
Differences in the relationship between landscape perception and physical and mental recovery across seasons. (**A**): Spring. (**B**): Autumn. Spring fitting indexes: Chi-square/df = 1.14, GFI = 0.94, AGFI = 0.93, RMSEA = 0.02, IFI = 0.99, NFI = 0.94, CFI = 0.99, TLI = 0.99, RFI = 0.94. Autumn fitting indexes: Chi-square/df = 1.18, GFI = 0.94, AGFI = 0.92, RMSEA = 0.03, IFI = 0.99, NFI = 0.94, CFI = 0.99, TLI = 0.99, RFI = 0.93. The two models fit well. * *p* < 0.05; ** *p* < 0.01; *** *p* < 0.001. SBP: systolic blood pressure; DBP: diastolic blood pressure. This figure was created by Rui Chen.

**Figure 8 ijerph-20-02357-f008:**
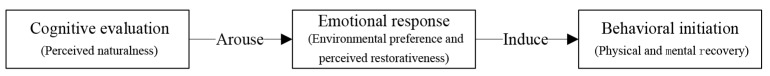
The process of human recovery from landscape space. This figure was created by Rui Chen.

**Figure 9 ijerph-20-02357-f009:**
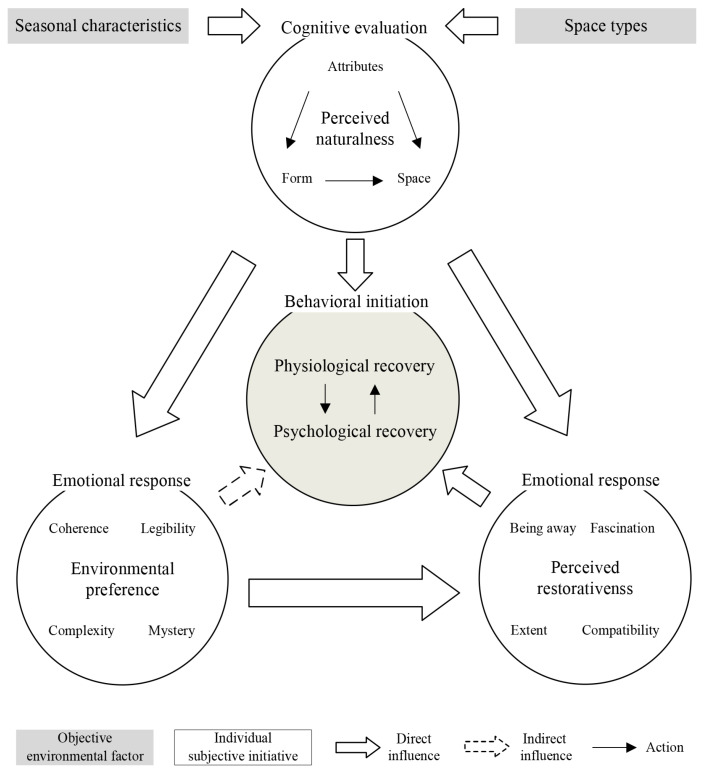
Effects of landscape perception on physical and mental recovery. This figure was created by Rui Chen.

**Table 1 ijerph-20-02357-t001:** Videos of six different types of forest landscapes.

Landscape Types	Spring	Autumn
Dynamic water	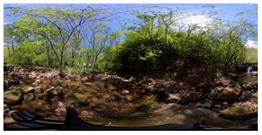	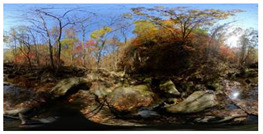
Static water	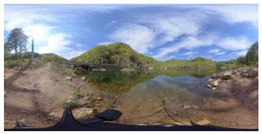	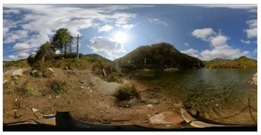
Lookout	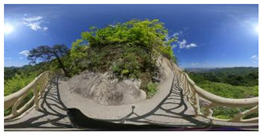	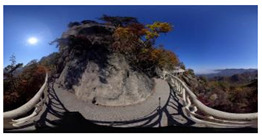
Coniferous forest	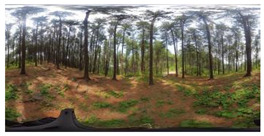	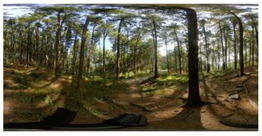
Broadleaved forest	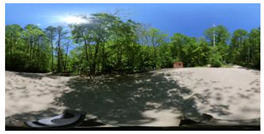	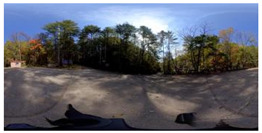
Mixed forest	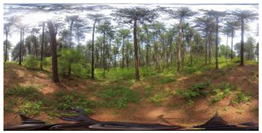	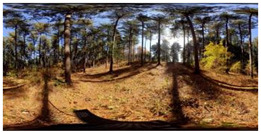

**Table 2 ijerph-20-02357-t002:** Perceived naturalness scale.

Evaluation Dimension	Evaluation Index			
Natural attributes	High vegetation coverage	Abundant plant species	Strong natural voice	Rich colors
Natural space	Feels friendly	Feels calm	Feels open	Feels wild
Natural form	Diverse vegetation contour	Terrain ups and downs	Vertical landscape hierarchy	Depth of landscape

**Table 3 ijerph-20-02357-t003:** Demographics of the participants (*n* = 520).

Landscape Types	Season	Male	Female	Total	Age
Dynamic water	Spring	21	22	43	18.93 ± 1.08
	Autumn	21	23	44	19.02 ± 1.17
Static water	Spring	21	23	44	18.86 ± 1.00
	Autumn	22	22	44	19.00 ± 1.18
Lookout	Spring	20	23	43	18.93 ± 1.10
	Autumn	21	22	43	18.88 ± 0.96
Coniferous forest	Spring	20	23	43	18.88 ± 1.16
	Autumn	21	22	43	19.60 ± 1.28
Broadleaved forest	Spring	21	22	43	18.88 ± 0.91
	Autumn	21	22	43	18.77 ± 1.00
Mixed forest	Spring	21	22	43	18.79 ± 1.15
	Autumn	21	23	44	18.68 ± 0.88

## Data Availability

The datasets analyzed during the current study are not publicly available due to continuing studies but are available from the corresponding author on reasonable request.

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
