# Peer review of "How Does the Experience of Forest Recreation Spaces in Different Seasons Affect the Physical and Mental Recovery of Users?"

_ijerph, 2023, doi:10.3390/ijerph20032357_

Round 1

Reviewer 1 Report

The findings are interesting and important and there is a lot in findings from analysis to discuss and describe.

I provide the pdf with comment, underlined and highlighted sentences, and added text.

English through the article needs to be simplified. Including abstract and introduction.

The discussion can present significant comparative findings more succinctly. I highlight comparative findings that seem interesting as examples. These findings and other could be more directly presented in discussion.

The findings and suggestions need to be separated. See suggestion in comment.

Author Response

Response letter

Dear reviewer,

Thanks for your valuable suggestions, we have made the following modifications according to your comments and suggestions.

In addition, our revised manuscript uses the "Track Changes" function. The specific modifications are as follows:

Comment 1:

The findings are interesting and important and there is a lot in findings from analysis to discuss and describe. I provide the pdf with comment, underlined and highlighted sentences, and added text.

Modify 1:

Thank you for your affirmation and the detailed comments on pdf.

Comment 2:

The discussion can present significant comparative findings more succinctly. I highlight comparative findings that seem interesting as examples. These findings and other could be more directly presented in discussion.

Modify 2:

Thank you for your suggestion. We have revised these sections based on your advice.

  • See as Line 458-459. The details are as follows:

Change “4.1.1 Effects of space types on landscape perception and physical and mental recovery”

to “4.1.1 Water and lookout landscapes have obtained higher perceptual evaluation and physical and mental recovery”

  • See as Line 487. The details are as follows:

Change “4.1.2 Effect of Landscape Perception on Physical and Mental Recovery”

to “4.1.2 Seasonal changes in FRSs can better promote perception and recovery”

  • See as Line 5 The details are as follows:

Change “4.2.1 Effects of seasonal characteristics on landscape perception and physical and mental recovery”

to “4.2.1 Perceptual evaluation of FRSs induces physical and mental recovery”

  • See as Line 557-5 The details are as follows:

Change “4.2.2 Effects of landscape perception on physical and mental recovery across seasons”

to “4.2.2 Effects of attributes and form of FRSs on physical and mental recovery (Spring: attributes+form; Autumn: attributes)”

Comment 3:

The findings and suggestions need to be separated. See suggestion in comment.

Modify 3:

Thanks for your review, and we have revised these sections based on your advice.

See as Line 645-646. The details are as follows:

Change “(1) Through the research, we not only found the important role of the natural attributes of forest space but also found that natural form plays a key role in promoting human health.”

to “(1) We found not only the important role of natural attributes of forest space but also that natural form is important for promoting human health.”

See as Line 653-657. The details are as follows:

Change “(2) When experiencing forest recreation space, meeting with insects or small animals may bring unexpected surprises to recreationists, thus enhancing their interest and producing a better recovery effect. Therefore, in the design of recreation space, the configuration of fruit-source plants and honey-source plants can be considered to improve the chance of meeting between recreationists and insects or small animals.”

to “(2) This study showed that meeting with insects or small animals may bring unexpected surprises to recreationists, thus enhancing their interest and producing a better recovery effect. Therefore, in the design of recreation space, the configuration of fruit-source plants and honey-source plants can be considered to improve insects and small animals presence for recreationists.”

See as Line 658-662. The details are as follows:

Change “(3) Considering the different restoration effects brought by the characteristic factors of the landscape environment, more forest rehabilitation activities can be planned in water landscapes and lookout landscapes. In addition, from the perspective of season, the forest therapy activities in the water landscapes and lookout landscapes in spring will produce a better recovery effect, and forest therapy activities in the in-forest landscapes in autumn will produce a better recovery effect.”

to “(3) This study also found that forest therapy activities in water and lookout landscapes in spring, and forest therapy activities in autumn will produce a better recovery effect. Considering the different restoration effects brought by the characteristic factors of the landscape environment, it is suggested to plan more forest rehabilitation activities in water and lookout landscapes.”

Comment 4:

English through the article needs to be simplified. Including abstract and introduction.

Modify 4:

Thank you for your comments, we highly cherish your comments. We have simplified the article appropriately.

In addition, we have made the following modifications according to your suggestions in the pdf annotations:

Comment 5:

Specify coniferous forest and long term preference or effect. And season as provided in following sections.

Modify 5:

Thank you for your suggestion. We highly cherish your comments. We added a description of the perceptual evaluation of coniferous and broadleaved forests to section “3.2.2 Seasonal differences in landscape perception across recreation spaces”.

See as Line 394-396.

“Furthermore, there was no significant seasonal difference (p > 0.05) between coniferous forest and broadleaved forest in each evaluation dimension (Figure 5).”

Comment 6:

Important for understanding findings: Dependent on the psychological state of each individual, the effect or response to natural environment will be more significant or not. Worse psychological state greater the effect.

Put in conclusion, or factors to consider for future studies.

Modify 6:

Thanks for your suggestion. According to your suggestion, we have modified this

part.

See as Line 484-486.

“It is worth noting that worse mental states may respond more significantly to the natural environment, and each individual mental states should be considered in future studies.”

Comment 7:

Important finding: Forward preference to season coming

Modify 7:

Thanks for your comment. We highlight this point in the text.

See as Line 503-504.

“This suggests a forward preference for the coming season.”

Comment 8:

Use different or combined instead mixed.

Modify 8:

Thank you for your suggestion. Following your suggestion, we changed the words in the manuscript.

See as Line 520-522. The details are as follows:

Change “At the same time, we found that insect elements such as butterflies and bees existed in the experimental video of the mixed forest landscape in spring, which made it receive positive evaluation.”

to “At the same time, we found that insect elements such as butterflies and bees existed in the experimental video of combined forest landscape in spring, which made it receive positive evaluation.”

Thanks again for your valuable feedback.

Yours sincerely,

Tong ZHANG and Rui CHEN

Reviewer 2 Report

In this manuscript, the Authors focused on the effect of forest recreation landscape spaces on the physical and mental recovery of users. The study fits well with the scope and aims of the Journal. The manuscript is well written and easy to read. The methodological approach is based on scientific literature and well-known statistical procedures (paired samples t test, one-way ANOVA and the independent-samples t t). The limitations of the study are clearly stated. The manuscript shows minor pitfalls. Please see the detailed report below. In my opinion, the manuscript requires minor revisions.

 Detailed report

 Lines 86-88: “The study showed that the higher participants’ evaluations of natural scenes were, the higher their perceived recovery and preference scores [19]”. Is the sentence referred to the previous one, i.e., “Based on the study of vegetation density, dense understory vegetation space is not popular [31], and people prefer space with a strong sense of privacy and security [32]”? It is unclear.

Lines 101-102: “The flowering stage is obviously the most popular”. Please consider avoiding personal opinions if they are not supported by adequate references.

Lines 165-167: “Considering the travel distance and time of people, this study selects the forest recreation areas (forest parks, scenic spots and scenic recreation forests) within the city suburbs (within 2 hours’ drive from the city center) as the research object”. As different distances can be traveled in 2 hours depending on the type of road, the vehicle used, the speed of travel, etc., I would suggest the Authors be more precise and state in place of time (2 hours) the distance in kilometers, if possible. Furthermore, please consider providing a definition of “time of people”.

Line 252: “Trier Social Stress Test”. Please consider adding adequate reference, i.e., references that support the use of the Trier Social Stress Test.

Figures 3, 4, and 5: in the legend, the colors of “Water landscapes”, “Lookout landscape”, and “In-forest landscapes” are too faded and almost imperceptible. Please consider making the colors more visible.

Lines 394-395: “in the four indexes of natural attributes(static water: t = -2.390, p = 0.0190), natural space(lookout: t = 2.268, p = 0.0259), legibility(static water: t = 2.116, p = 0.0373)”. Please fix the typos.

Figure 6 is unclear. The Authors could insert the figure vertically to allow enlargements of the elements and text.

Figure 7 is unclear. The Authors could place the two figures (Spring and Autumn) vertically aligned on top of each other.

Figure 9 could be made more readable.

Figure A1 is unclear. The Authors could split up the figure into twelve figures and place each figure on a single page.

Author Response

Response letter

Dear reviewer,

Thanks for your valuable suggestions, we have made the following modifications according to your comments and suggestions.

In addition, our revised manuscript uses the "Track Changes" function. The specific modifications are as follows:

Comment 1:

Lines 86-88: “The study showed that the higher participants’ evaluations of natural scenes were, the higher their perceived recovery and preference scores [19]”. Is the sentence referred to the previous one, i.e., “Based on the study of vegetation density, dense understory vegetation space is not popular [31], and people prefer space with a strong sense of privacy and security [32]”? It is unclear.

Modify 1:

Thanks for your advice. We value your advice very much and we are sorry that we did not express it clearly. “The study showed that the higher participants’ evaluations of natural scenes were, the higher their perceived recovery and preference scores [19].” The sentence does not refer to the previous one, but a new study finding. At the same time, the relevant expressions are modified.

See as Line 86-88. The details are as follows:

Change “The study showed that the higher participants’ evaluations of natural scenes were, the higher their perceived recovery and preference scores [19]”

to “Another study by Carrus et al. showed that the higher participants’ evaluations of natural scenes were, the higher their perceived recovery and preference scores [19].”

Comment 2:

Lines 101-102: “The flowering stage is obviously the most popular”. Please consider avoiding personal opinions if they are not supported by adequate references.

Modify 2:

Thank you for your suggestion. “The flowering stage is obviously the most popular.” The sentence belongs to the same reference as the following sentence “In addition, the color change in deciduous trees and bushes in autumn is also highly preferred [40].” And we have modified the expression of the two sentences.

See as Line 101-103. The details are as follows:

Change “The flowering stage is obviously the most popular. In addition, the color change in deciduous trees and bushes in autumn is also highly preferred [40].”

to “The flowering stage is obviously the most popular, and the color change in deciduous trees and bushes in autumn is also highly preferred [40].”

Comment 3:

Lines 165-167: “Considering the travel distance and time of people, this study selects the forest recreation areas (forest parks, scenic spots and scenic recreation forests) within the city suburbs (within 2 hours’ drive from the city center) as the research object”. As different distances can be traveled in 2 hours depending on the type of road, the vehicle used, the speed of travel, etc., I would suggest the Authors be more precise and state in place of time (2 hours) the distance in kilometers, if possible. Furthermore, please consider providing a definition of “time of people”.

Modify 3:

Thank you for your suggestion. According to your suggestion, we revised what you said in the manuscript.

See as Line 165-167. The details are as follows:

Change “Considering the travel distance and time of people, this study selects the forest recreation areas (forest parks, scenic spots and scenic recreation forests) within the city suburbs (within 2 hours’ drive from the city center) as the research object.”

to “Considering the travel distance of people, this study selects the forest recreation areas (forest parks, scenic spots and scenic recreation forests) within the city suburbs (within 60-100 km from the city center) as the research object. ”

Comment 4:

Line 252: “Trier Social Stress Test”. Please consider adding adequate reference, i.e., references that support the use of the Trier Social Stress Test.

Modify 4:

Thanks for your review, and we highly cherish your comments.

According to your comments, we have cited three papers to illustrate the feasibility of the Trier Social Stress Test used in this study.

See as Line 253-254.

“In this study, the TSST consisted of a 3-minute public presentation and a 2-minute oral arithmetic task under noise [15,50,53].”

Linking to the literature:

  1. Wang, X.B.; Shi, Y.X.; Zhang, B.; Chiang, Y.C. The Influence of Forest Resting Environments on Stress Using Virtual Reality. J. Environ. Res. Public. Health2019, 16, 3263, doi:10.3390/ijerph16183263.
  2. Jiang, B.; Chang, C.Y.; Sullivan, W.C. A Dose of Nature: Tree Cover, Stress Reduction, and Gender Differences. Urban Plan.2014, 132, 26–36, doi:10.1016/j.landurbplan.2014.08.005.
  3. Annerstedt, M.; Jönsson, P.; Wallergård, M.; Johansson, G.; Karlson, B.; Grahn, P.; Hansen, Å.M.; Währborg, P. Inducing Physiological Stress Recovery with Sounds of Nature in a Virtual Reality Forest — Results from a Pilot Study. Behav.2013, 118, 240–250, doi:10.1016/j.physbeh.2013.05.023.

Comment 5:

Figures 3, 4, and 5: in the legend, the colors of “Water landscapes”, “Lookout landscape”, and “In-forest landscapes” are too faded and almost imperceptible. Please consider making the colors more visible.

Modify 5:

Thanks for your suggestion, which is highly valued. We changed the colors of “Water landscapes”, “Lookout landscape” and “In-forest landscapes” to make them more obvious.

Figure 3 see as Line 324.

Figure 4 see as Line 354.

Figure 5 see as Line 366.

Comment 6:

Lines 394-395: “in the four indexes of natural attributes(static water: t = -2.390, p = 0.0190), natural space(lookout: t = 2.268, p = 0.0259), legibility(static water: t = 2.116, p = 0.0373)”. Please fix the typos.

dimensions

Modify 6:

Thanks for your review, and we highly value your comments.

We carefully checked the text and Figure 5 and found that water and lookout landscapes had seasonal differences in the four dimensions: “natural attributes”, “natural space”, “legibility” and “extent”. And also revised the manuscript.

See as Line 391-394. The details are as follows:

Change “This is reflected in the four indexes of natural attributes(static water: t = -2.390, p = 0.0190), natural space(lookout: t = 2.268, p = 0.0259), legibility(static water: t = 2.116, p = 0.0373) and extent (dynamic water: t = -2.477, p = 0.0152; lookout: t = 3.188, p = 0.0020).”

to “This is reflected in the four dimensions of natural attributes(static water: t = -2.390, p = 0.0190), natural space(lookout: t = 2.268, p = 0.0259), legibility(static water: t = 2.116, p = 0.0373) and extent (dynamic water: t = -2.477, p = 0.0152; lookout: t = 3.188, p = 0.0020).”

Comment 7:

Figure 6 is unclear. The Authors could insert the figure vertically to allow enlargements of the elements and text.

Modify 7:

Thank you for your advice. We put it alone on horizontal paper to enlarge the elements and text.

See as Line 417.

Comment 8:

Figure 7 is unclear. The Authors could place the two figures (Spring and Autumn) vertically aligned on top of each other.

Modify 8:

Thanks for your comment. According to your suggestion, we aligned the two graphs vertically together.

See as Line 427.

Comment 9:

Figure 9 could be made more readable.

Modify 9:

Thank you for your suggestion. We highly cherish your comments.

In the figure, we abbreviated the three evaluation dimensions of perceived naturalness, and add a background color to the final link of the influence mechanism “physical and mental recovery” to make it more readable.

See as Line 602.

Comment 10:

Figure A1 is unclear. The Authors could split up the figure into twelve figures and place each figure on a single page

Modify 10:

Thank you for your suggestion. According to your suggestion, we have modified this part. Considering the length and content of the article, we split up the figure into two pictures (spring and autumn) from the perspective of the season and place each figure on a single page.

Figure A1 see as Line 684.

Figure A2 see as Line 690.

Thanks again for your valuable feedback.

Yours sincerely,

Tong ZHANG and Rui CHEN
